# Hypertension screening across different age groups in Indian adults: Evidence from nationally representative cross-sectional data

**Prakash Mathiyalagen[1], Anand Rajagopal[1], Kavita Vasudevan[1], Sridevi Gnanasekaran[1], Sharath Burugina Nagaraja[2], Jayanta Bora[3], Sonu Goel[4,\*]**

**1** Dept of Community Medicine, IGMC&RI, Puducherry, India, **2** ESIC Medical College and PGIMSR, Bengaluru, Karnataka, India, **3** VART Consulting Pvt, Ltd, Delhi-NCR & Mumbai, India, **4** Postgraduate Institute of Medical Education and Research (PGIMER), Chandigarh, India

\* sonugoel007@yahoo.co.in

## Abstract

### Background

Hypertension is a significant public health concern globally and in India, contributing substantially to the burden of cardiovascular diseases. Early detection through appropriate screening practices is critical to mitigating its long-term impact. However, there is limited evidence on age-specific screening practices for hypertension in the Indian population. This study aims to assess hypertension screening practices across different age groups and identify associated socio-demographic, behavioral, and physiological factors to inform effective public health strategies.

### Methods

A cross-sectional study was conducted using data from the fifth National Family Health Survey (NFHS-5), which included 1,01,839 men and 7,24,115 women aged 15 years and above. Hypertension was defined based on systolic blood pressure (SBP) ≥140 mmHg and/or diastolic blood pressure (DBP) ≥90 mmHg. Multivariate logistic regression was used to analyze associations between hypertension and various socio-demographic and behavioral factors. Receiver Operating Characteristic (ROC) analysis determined the optimal age for initiating hypertension screening.

### Results

The prevalence of hypertension increased linearly with age, with a significant rise observed in the 35-39 years age group. Factors significantly associated with hypertension included marital status, religion, education, wealth index, alcohol consumption, and waist-hip ratio. ROC analysis identified 35 years as the optimal age for initiating routine hypertension screening in both men and women. Screening in this age group showed balanced sensitivity and specificity for early detection of hypertension.

**Data availability statement:** This study is based on data from the Demographic and Health Surveys (DHS) Program (https://dhsprogram.com/). At the time of article publication, DHS data are no longer accessible due to a pause in operations resulting from an ongoing review of U.S. foreign assistance programs. The authors did not receive special access and obtained the data through standard procedures available to all registered researchers at that time.

**Funding:** The author(s) received no specific funding for this work.

**Competing interests:** none.

## Conclusions

This study highlights the need to revise hypertension screening policies in India, recommending initiation at 35 years to improve early detection and management. Addressing socio-demographic and lifestyle determinants through targeted interventions is critical to achieving national hypertension control goals.

## Introduction

Worldwide, the prevalence of hypertension among adults (30–79 years) in 2019 was 32% in women and 34% in men [1]. Among those with hypertension, 41% of women and 51% of men did not have a previous diagnosis of hypertension [1]. Less than half of the people with hypertension had taken treatment leading to the treatment rate of 47% in women and 38% in men [1]. The hypertension control after taking treatment was achieved for 23% women and 18%men with hypertension [1].As per National Non-communicable Disease Monitoring Survey (NNMS) in India, the prevalence of hypertension among adults aged 18–69 years was 28.5%. Among them, 27% were aware of their hypertensive status, 14.5% were on treatment for hypertension and 12.6% had blood pressure under control [2].

Globally in 2019, high systolic blood pressure (SBP) was the leading risk factor for attributable deaths, accounting for 10.8 million deaths (19.2% of all deaths in 2019) [3]. High SBP was attributed to 5.25 million (20.3%) and 5.60 million (18.2%) deaths among females and males respectively in 2019 [3]. The leading risk factor in 2019 by percentage of Disability Adjusted Life Years (DALYs) was attributed to high SBP with 9.3 (8.2 to 10.5), 6.0 (4.9 to 7.1), 16.1 (14.2 to 18.0) and 19.5 (16.3 to 22.7) for all ages, 25–49 years, 50–74 years and 75 years and above respectively [3]. In most parts of southeast Asia, 10% to 20% of DALYs were attributable to high SBP [3].

International Society of Hypertension (ISH) recommended that for those with normal blood pressure (BP) level (<130/85 mmHg), BP would be re-measured after 1 year and 3 years in those with risk factors and without risk factors respectively [4]. US Preventive Services Task Force (USPSTF) acknowledged the lack of evidence for optimal screening intervals and recommended the screening practice of once a year among those adults aged 40 and above and also in adults with risk factors and every 3–5 years among adults aged 18–39 years without any risk factors for hypertension [5]. While several LMICs face a growing burden of hypertension, few have conducted large-scale, nationally representative studies to determine an age threshold for screening. A study in Nepal by Dhungana et al. (2022) noted that younger adults aged 15–29 years were significantly less likely to have been screened for hypertension, highlighting a missed opportunity for early detection in LMICs [6]. However, this study did not provide a precise age cutoff for initiating screening. Our study is among the first in India and similar LMIC settings to use ROC analysis to empirically determine an optimal screening age.

In India, the recommendation under standard treatment guidelines for hypertension was to re-check BP in 2 years among persons with normal BP (<130/80 mmHg)

and in 1 year for persons with high normal BP [7]. Under the National Programme for Prevention and Control of Cancer, DM, Cardiovascular Diseases and Stroke (NPCDCS), the recommended practice was to screen once a year for aged 30 years and above [8]. Despite India's recommendation of initiating hypertension screening at 30 years under NPCDCS, there is no age-stratified evidence to support this threshold from nationally representative datasets. There is a need for sensitizing the health care providers to screen for high BP at optimum intervals without the need for unnecessary screening and overburdening them [9].

Among hypertensive patients, only 76% had ever received a BP screening according to NFHS-4 [10] and 64% had not checked their BP in the last one year in Puducherry, India [11]. In Nepal, the prevalence of hypertension screening was 65.9% among the hypertensive individuals [6]. Hypertension screening was significantly lower among those in the age group of 15–29 years (52.2%), those who were never married (54.4%) and those with no formal schooling (59.6%) [6]. In Germany, 37.5% of men and 21.9% of women did not have BP measurement within the last one year among non hypertensive individuals [12]. In Canada, 2.6% of the individuals did not have BP measurement ever and 9.0% did not have their BP assessed in the last 2 years [13]. In America, 87.4% of individuals of aged 18–39 years have never measured their BP at home [12]. Even though there is a policy for regular BP measurement with every country [12], there is a gap in the implementation of these policies.

Evidence suggests that SBP increases along with age in a linear relationship whereas DBP increases before 45 years of age and then decreases afterwards in a curvilinear fashion [14]. However the relationship between and age varies from one country to another depending on the contributing factors [14]. Hence there is a need to identify the relationship between age and BP and to find the appropriate age at which the screening is required for early diagnosis.

The Government of India has adopted a national multi-sectoral action plan for the prevention and control of non-communicable diseases and set a target of 25% relative reduction in the prevalence of raised blood pressure by 2025 [15]. Since hypertension is an asymptomatic condition in most of the individuals, it can increase the overall risk of cardiovascular diseases (CVD) and lead to life threatening complications like heart attack, stroke and renal failure [7]. Early diagnosis of high BP is achieved by appropriately screening the people at adequate intervals. Screening for high BP at an appropriate interval and at an appropriate age will not only help the individuals to lower the CVD risk factors but also in attaining the Government's target towards reducing the high BP prevalence in the entire community.

There is a paucity of evidence to suggest the appropriate screening practice for a particular age group from a nationwide representative sample, although there are some recommendations existing depending on the blood pressure values and for all age groups in general. In this regard, the present study has been planned with the objective of determining the hypertension screening practice across different age groups among the Indian population from a nationwide representative sample (NFHS-5).The objectives of the current study were to assess the hypertension screening practice across different age groups in Indian adults and to identify the factors associated with hypertension among Indian adults.

## Materials and methods

### Study design

This cross sectional study utilizes secondary data from the fifth National Family Health Survey (NFHS-5), which is the Indian version of the Demographic and Health Survey conducted regularly in more than 90 countries worldwide. NFHS is a collaborative project supported by the International Institute of Population Sciences (IIPS) in Mumbai, India; International Classification of Functioning, Disability and Health (ICF) in Calverton, Maryland, USA, and East West Center in Honolulu, Hawaii, USA. The Ministry of Health and Family Welfare (MoHFW) of the Government of India has appointed IIPS as the coordinating and technical guidance agency of NFHS. The survey is funded by the United States Agency for International Development (USAID) with additional support from the United Nations Children's Fund (UNICEF). NFHS uses four different types of questionnaires such as household, man's, woman's and biomarker in 19 languages using the Computer Assisted Personal Interviewing (CAPI). The fieldwork for NFHS-5 in India was done in two phases. Phase one took place

between June 17, 2019, and January 30, 2020, which covered 17 states and 5 Union Territories. Phase two was conducted between January 2, 2020 and April 30, 2021, which covered 11 states and 3 Union Territories. A total of 17 Field Agencies gathered data from 6,36,699 households, 1,01,839 men and 7,24,115 women.

## Data source

The current data analysis uses the men's questionnaire and women's questionnaire from NFHS- 5, mainly focusing on male and female datasets. It focuses on the various independent factors associated with hypertension (dependent variable) among Indian adults.Analyzing the male and female factors separately can provide a clear understanding on the social and cultural factors associated with hypertension. Potential confounding variables such as dietary sodium intake, psychological stress, and family history of hypertension were not included due to limitations in the NFHS-5 data collection instruments. It examines the ideal age points at which the Indian adult population has to be screened for hypertension. Usually, the age cut off for screening hypertension has been taken as 30 years and the evidence provided through a nationwide survey would be more convincing regarding the appropriate age cut off. The total sample size utilized for this secondary data analysis was 1,01,839 men and 7,24,115 women aged 15 years and above. Since there are missing values, each variable in this analysis has a different sample size, as mentioned in the results section.

## Ethical approval

The research was conducted using a pre-existing, anonymised data from a public source, which did not require direct involvement of the patients or the general public. Since the dataset did not contain any personally identifiable information about the study participants, a formal ethics committee approval was not mandatory. However, the ethics committee approval was obtained from the Institutional Ethics Committee of the Post Graduate Institute of Medical Education and Research (PGIMER) in Chandigarh (approval number IEC-08/2022–2535, dated 17.08.2022).

## Definitions

As per the guidelines of World Health Organisation (WHO) and American Heart Association (AHA), the study describes elevated blood pressure as having a systolic blood pressure of 140 mmHg or higher and/or a diastolic blood pressure of 90 mmHg or more [16]. The raised risk of metabolic complications was considered when the waist-hip ratio (WHR) was 0.90 or higher for men and 0.85 or higher for women [17]. Regions were divided into 6 categories namely north (Chandigarh, Delhi, Haryana, Himachal Pradesh, Jammu & Kashmir, Punjab, Rajasthan, Uttarakhand), central (Chattisgarh, Madhya Pradesh, Uttar Pradesh), east (Bihar, Jharkhand, Odisha and West Bengal), northeast (Arunachal Pradesh, Assam, Manipur, Meghalaya, Mizoram, Nagaland, Sikkim and Tripura), west (Dadra & Nagar Haveli, Daman & Diu, Goa, Gujarat, Maharashtra) and south (Andaman & Nicobar Islands, Andhra Pradesh, Karnataka, Kerala, Lakshadweep, Puducherry, Tamil Nadu and Telangana).

## Statistical analysis

Hypertension was identified as the outcome variable in the study. The average of three blood pressure readings that were collected during the NFHS was considered for the study. The quantitative blood pressure values were then converted into a dichotomous variable as presence or absence of hypertension. Sampling weights were incorporated into the data to address the differential probabilities of selection and participation.

Statistical analysis was performed using IBM SPSS version 20. The data were presented in the form of mean with 95% confidence interval (CI), frequency and percentage. The associations were evaluated using multivariable logistic regression analysis to calculate the adjusted odds ratio and 95% CI of raised blood pressure with age category (8 categories), marital status, religion, educational status, wealth index (5 categories), occupation, residence, region (6 categories), caste, waist-hip ratio, alcohol use, tobacco use, fruit intake, fried food intake and co-morbidities. The fitness of the dataset

was assessed using the Hosmer and Lemeshow test and the Goodness of Fit test. Receiver Operating Characteristics (ROC) Curve was used to evaluate and compare the performance of classification models (age) where the outcome variable is binary (raised blood pressure: present/absent). Sensitivity and specificity was used to describe the diagnostic ability of the screening test. The Youden index was used as a summary measure of the ROC curve to select the optimal cut off point for the age within the age group.A p value of less than 0.05 was considered as significant. Missing data were handled by listwise deletion. Participants with missing values for outcome or explanatory variables were excluded from respective analyses. No imputation was performed. Sample sizes for each analysis are reported in the tables to reflect variations due to missing data.

## Results

A total of 1,01,839 men and 7,24,115 women participants were included in the NFHS-5 study. Since the missing values were excluded from the study, the sample size varied across different variables. Tables 1 and 2 highlight the key findings from adult male population. The systolic and diastolic blood pressure increases with increasing age as shown in Figs 1 and 2, which is also evident from the statistically significant trends in the increasing odds ratio. Though SBP does not have any association with marital status, the DBP is showing a significant association with marital status. Married men are having an increased risk of 1.12 times for developing raised DBP compared to never married men. Muslim men are 10% and 14% lesser risk of developing raised SBP and DBP respectively, compared Hindu men. Christian men are 11% lesser risk of developing raised DBP whereas Others category (Sikh, Buddhist, Jain and Jewish) men are having 1.29 times higher risk of developing raised SBP compared to Hindu men. With an increase in educational status, the odds of developing raised SBP and DBP also increases. When compared to no education men, the primary, secondary and higher educated men show 1.14,1.20 and 1.01 times of having higher SBP. The primary, secondary and higher educated men show 1.09, 1.24 and 1.12 times of having higher DBP in comparison with no education men. There is a linear trend in wealth index for having raised SBP and DBP. Richer and richest quintile shows 1.12 and 1.14 times of higher risk of developing raised SBP compared to poorest quintile. Poorer, middle, richer and richest quintile shows 1.17, 1.32, 1.32 and 1.35 times respectively of developing raised DBP compared to poorest quintile. Though the employed men has 5% and 3% lesser risk of developing raised SBP and DBP, the association was not found to be statistically significant. The rural men have 8% lower risk of developing raised SBP and DBP compared to urban men. Men belonging to east zone have 34% and 33% lesser risk of developing raised SBP and DBP respectively, compared to men from north zone. West zone has 25% lesser risk of raised SBP whereas south zone has 1.27 times higher risk of raised DBP. Men belonging to ST caste have 1.15 times higher risk developing raised SBP whereas OBC and Others has 11% and 9% lesser risk of developing raised DBP respectively.

The men having WHR more than 0.90 have 1.46 and 1.62 times higher risk of developing raised SBP and DBP respectively. Men having a history of alcohol consumption have 1.28 and 1.50 times higher risk of developing raised SBP and DBP respectively. Men having a history of tobacco use 1.20 and 1.05 times higher risk of developing raised SBP and DBP respectively, though the association between raised DBP and tobacco use was not significant. Regarding fruit intake, daily, weekly and occasionally consuming men have 20%. 9% and 15% reduced risk of developing raised DBP respectively, compared to never category men. Regarding fried food intake, daily consuming men have 1.18 times increased risk of developing raised SBP, compared to never category men. Daily, weekly and occasional consumers of fried food have 10%, 9% and 7% reduced risk of developing raised DBP respectively, compared to never category men. There is no association observed between men with co-morbidities such as diabetes and cardio-vascular diseases and raised SBP and DBP.

Tables 3 and 4 highlight the key findings from adult female population. The systolic and diastolic blood pressure of women increases with increasing age as shown in Figs 3 and 4, which is also evident from the statistically significant trends in the increasing odds ratio. Though DBP does not have any association with marital status, the SBP is showing a

**Table 1. Distribution of systolic and diastolic blood pressure among males according to socio-demographic, behavioral and physiological characteristics of the study population, NFHS-5, 2019−21, India (N = 1,01,839).**

| Characteristics | N | (%) | Systolic blood pressure | | Diastolic blood pressure | |
|---|---|---|---|---|---|---|
| | | | Weighted mean | 95%CI | Weighted mean | 95%CI |
| **Socio-demography** | | | | | | |
| **Age category (in years)** | | | | | | |
| 15–19 | 14337 | 14.1 | 115.09 | 114.91-115.28 | 72.86 | 72.72-73.01 |
| 20–24 | 12473 | 12.2 | 119.14 | 118.95-119.34 | 76.53 | 76.37-76.68 |
| 25–29 | 12410 | 12.2 | 120.68 | 120.49-120.88 | 79.04 | 78.89-79.19 |
| 30–34 | 11592 | 11.4 | 122.63 | 122.41-122.84 | 81.00 | 80.84-81.17 |
| 35–39 | 11641 | 11.4 | 123.55 | 123.31-123.78 | 82.25 | 82.08-82.43 |
| 40–44 | 9667 | 9.5 | 124.96 | 124.66-125.25 | 83.52 | 83.31-83.72 |
| 45–49 | 10058 | 9.9 | 126.79 | 126.47-127.10 | 84.10 | 83.89-84.30 |
| 50–54 | 7947 | 7.8 | 128.67 | 128.30-129.03 | 84.57 | 84.34-84.80 |
| **Marital status** | | | | | | |
| Never married | 31756 | 31.2 | 118.00 | 117.87-118.13 | 75.75 | 75.65-75.85 |
| Currently married | 57031 | 56.0 | 124.22 | 124.11-124.34 | 82.16 | 82.08-82.24 |
| Widowed/Separated/Divorced | 1337 | 1.3 | 125.24 | 124.42-126.06 | 82.65 | 82.11-83.19 |
| **Religion** | | | | | | |
| Hindu | 72249 | 70.9 | 122.10 | 122.00-122.20 | 80.03 | 79.95-80.10 |
| Muslim | 13122 | 12.9 | 121.16 | 120.94-121.39 | 78.83 | 78.67-79.00 |
| Christian | 2484 | 2.4 | 123.22 | 122.68-123.76 | 80.87 | 80.48-81.26 |
| Others* | 2269 | 2.2 | 124.09 | 123.48-124.69 | 81.27 | 80.83-81.70 |
| **Educational status** | | | | | | |
| No education | 10893 | 10.7 | 122.95 | 122.68-123.22 | 80.93 | 80.73-81.12 |
| Primary | 10990 | 10.8 | 122.83 | 122.57-123.10 | 80.65 | 80.46-80.84 |
| Secondary | 51460 | 50.5 | 121.58 | 121.46-121.70 | 79.45 | 79.36-79.54 |
| Higher | 16781 | 16.5 | 122.38 | 122.18-122.57 | 80.17 | 80.03-80.32 |
| **Wealth Index of the household** | | | | | | |
| Poorest (I quintile) | 15204 | 14.9 | 120.28 | 120.07-120.49 | 78.18 | 78.02-78.33 |
| Poorer (II quintile) | 17872 | 17.5 | 121.21 | 121.00-121.41 | 79.16 | 79.01-79.31 |
| Middle (III quintile) | 19497 | 19.1 | 122.03 | 121.83-122.22 | 80.08 | 79.94-80.23 |
| Richer (IV quintile) | 20190 | 19.8 | 122.76 | 122.57-122.95 | 80.67 | 80.53-80.81 |
| Richest (V quintile) | 17362 | 17.0 | 123.64 | 123.43-123.84 | 81.12 | 80.96-81.27 |
| **Current occupation** | | | | | | |
| Unemployed | 21208 | 20.8 | 118.27 | 118.10-118.45 | 76.18 | 76.05-76.31 |
| Employed | 68917 | 67.7 | 123.20 | 123.10-123.31 | 81.06 | 80.98-81.13 |
| **Residence** | | | | | | |
| Urban | 30569 | 30.0 | 122.84 | 122.68-122.99 | 80.66 | 80.55-80.77 |
| Rural | 59555 | 58.5 | 121.64 | 121.53-121.75 | 79.52 | 79.44-79.61 |
| **Region** | | | | | | |
| North | 7722 | 7.6 | 124.57 | 124.28-124.87 | 80.90 | 80.68-81.11 |
| Central | 10302 | 10.1 | 123.21 | 122.95-123.46 | 80.31 | 80.12-80.51 |
| East | 23013 | 22.6 | 120.36 | 120.19-120.53 | 78.28 | 78.15-78.40 |
| Northeast | 5000 | 4.9 | 122.87 | 122.49-123.25 | 80.16 | 79.88-80.43 |
| West | 21571 | 21.2 | 121.28 | 121.10-121.46 | 79.70 | 79.56-79.83 |
| South | 22516 | 22.1 | 122.93 | 122.74-123.12 | 81.20 | 81.06-81.35 |

*(Continued)*

Table 1. (Continued)

| Characteristics | N | (%) | Systolic blood pressure | | Diastolic blood pressure | |
|---|---|---|---|---|---|---|
| | | | Weighted mean | 95%CI | Weighted mean | 95%CI |
| **Caste** | | | | | | |
| SC | 18473 | 18.1 | 122.04 | 121.83-122.23 | 80.05 | 79.90-80.20 |
| ST | 8366 | 8.2 | 122.37 | 122.07-122.66 | 80.01 | 79.79-80.23 |
| OBC | 38188 | 37.5 | 121.78 | 121.64-121.92 | 79.82 | 79.71-79.92 |
| Others | 25096 | 24.6 | 122.35 | 122.18-122.52 | 79.91 | 79.78-80.03 |
| **Behavioral and Physiological** | | | | | | |
| **Waist Hip Ratio category** | | | | | | |
| >/=0.90 | 44168 | 43.4 | 124.20 | 124.07-124.33 | 82.01 | 81.91-82.11 |
| <0.90 | 40523 | 39.8 | 119.73 | 119.61-119.86 | 77.67 | 77.57-77.76 |
| **History of Alcohol use** | | | | | | |
| No | 69279 | 68.0 | 121.31 | 121.21-121.41 | 79.12 | 79.04-79.19 |
| Yes | 20845 | 20.5 | 124.49 | 124.30-124.69 | 82.54 | 82.39-82.69 |
| **History of tobacco use** | | | | | | |
| No | 78259 | 76.8 | 121.92 | 121.82-122.02 | 79.74 | 79.67-79.81 |
| Yes | 11866 | 11.7 | 122.87 | 122.62-123.12 | 81.01 | 80.82-81.19 |
| **Frequency of fruit intake** | | | | | | |
| Never | 1490 | 1.5 | 121.10 | 120.37-121.83 | 80.12 | 79.58-80.67 |
| Daily | 10590 | 10.4 | 122.72 | 122.46-122.98 | 80.28 | 80.08-80.48 |
| Weekly | 39478 | 38.8 | 122.38 | 122.25-122.52 | 80.21 | 80.11-80.31 |
| Occasionally | 38567 | 37.9 | 121.55 | 121.41-121.68 | 79.49 | 79.39-79.59 |
| **Frequency of fried food intake** | | | | | | |
| Never | 6883 | 6.8 | 122.53 | 122.19-122.86 | 80.68 | 80.44-80.93 |
| Daily | 8018 | 7.9 | 122.46 | 122.15-122.76 | 79.72 | 79.50-79.94 |
| Weekly | 32160 | 31.6 | 121.67 | 121.53-121.82 | 79.62 | 79.51-79.73 |
| Occasionally | 43063 | 42.3 | 122.17 | 122.04-122.30 | 80.04 | 79.94-80.14 |
| **Co-morbidities** | | | | | | |
| Hypertension | 3078 | 3.0 | 132.26 | 131.57-132.94 | 87.58 | 87.12-88.04 |
| Diabetes | 2497 | 2.5 | 127.16 | 126.54-127.78 | 84.24 | 83.82-84.66 |
| Cardiovascular disease | 956 | 0.9 | 122.92 | 121.98-123.86 | 81.41 | 80.76-82.05 |

*Includes Sikh, Buddhist, Jain, Jewish and others.

significant association with marital status. Married women are having 21% reduced risk of developing raised SBP compared to never married women. Muslim women are 1.13 and 1.16times increased risk of developing raised SBP and DBP respectively, compared Hindu women. Others category (Sikh, Buddhist, Jain and Jewish) women are having 1.45 and 1.32 times higher risk of developing raised SBP and DBP respectively, compared to Hindu women. With an increase in educational status, the odds of developing raised SBP and DBP decreases. When compared to no education women, the primary, secondary and higher educated women show 9%, 8% and 28% reduced risk having higher SBP. The higher educated women show 20%reduced risk having higher DBP in comparison with no education women. Middle, richer and richest quintile shows 1.13, 1.25 and 1.23 times of higher risk of developing raised DBP compared to poorest quintile. However, there was no association observed between SBP and wealth index among women. The employed women are 14% and 5% lesser risk of developing raised SBP and DBP respectively, compared to unemployed women. There was no association observed between raised BP and residence among women. Women belonging to central, north-east and south zone have 1.29, 1.28 and 1.20 times increased risk of developing raised SBP respectively, compared to women from north zone. Similarly,

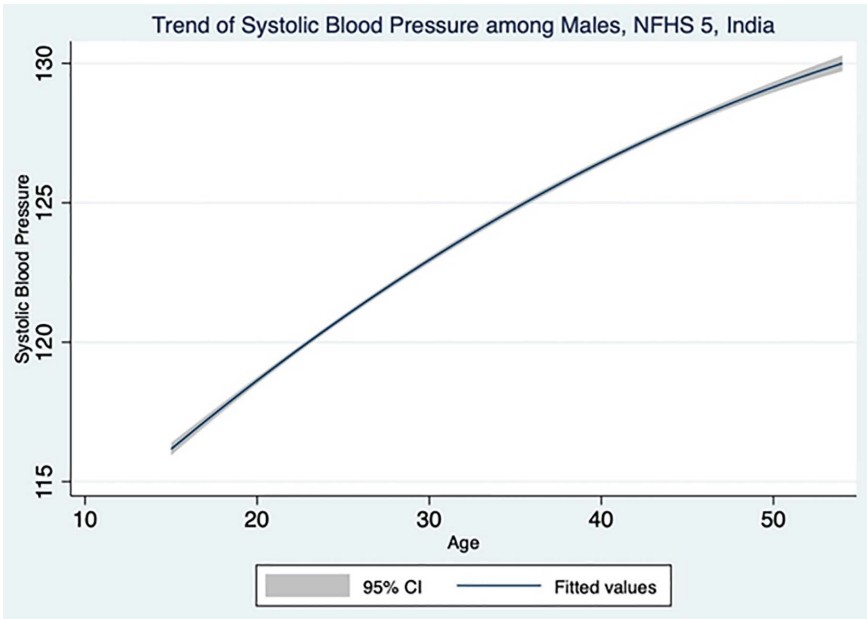

**Fig 1. Male systolic blood pressure across different age group in the study population.**

women belonging to central, north-east and south zone have 1.22, 1.17 and 1.32 times increased risk of developing raised DBP respectively. Women belonging to ST caste have 1.27 and 1.16 times higher risk developing raised SBP and DBP respectively, whereas women belonging to OBC have 8% lesser risk of developing raised DBP.

The women having WHR more than 0.85 have 1.38 and 1.38 times higher risk of developing raised SBP and DBP respectively. Women having a history of alcohol consumption have 1.44 and 1.56 times higher risk of developing raised SBP and DBP respectively. There was no association observed between raised BP and tobacco use among women. There was no association observed between raised BP and fruit intake among women. Regarding fruit intake, daily, weekly and occasionally consuming men have 20%. 9% and 15% reduced risk of developing raised DBP respectively, compared to never category men. Regarding fried food intake, weekly and occasional consuming women have 24% and 19% reduced risk of developing raised SBP, compared to never category women. Daily, weekly and occasional consumers of fried food have 13%, 17% and 14% reduced risk of developing raised DBP respectively, compared to never category women. Women with diabetes have 1.53 and 1.60 times increased risk of developing raised SBP and DBP respectively. Table 5, Figs 5 and 6 highlights the cut off age in years for adult men using Receiver Operating Characteristics (ROC) Curve. The ROC has been used to evaluate and compare the performance of age as the screening tool to identify the binary outcome variable (raised blood pressure). The Youden index was used as a summary measure of the ROC curve to select the optimal cut off point for the age within the age group. The ROC analysis yielded an AUC of 0.71 (95% CI: 0.70–0.72) for males and 0.68 (95% CI: 0.67–0.69) for females when using age to predict elevated systolic blood pressure (SBP). The 35–39 years age group has shown the least difference between sensitivity and specificity and hence it can be suggested as the ideal screening age among adult men for raised SBP and DBP. Table 6, Figs 7 and 8 highlights the cut off age in years for adult women using ROC Curve. Similar to adult men, the 35–39 years age group has shown the least difference between sensitivity and specificity and hence it can be suggested as the ideal screening age among adult women for raised SBP and DBP. The Youden index identified 36.5 years as the age cutoff with the best balance of sensitivity (57.9%) and specificity (49.7%) for men, and similar findings for women, supporting the recommendation of 35 years as a screening initiation point.

**Table 2. Multivariate logistic regression analysis to determine the factors associated with elevated systolic and diastolic blood pressure among male, NFHS-5, 2019−21, India.**

| Characteristics | SBP (>/=140 mmHg) | | | DBP (>/=90 mmHg) | | |
|---|---|---|---|---|---|---|
| | OR | 95% CI | | OR | 95% CI | |
| | | Lower | Upper | | Lower | Upper |
| **Socio-demographic characteristics** | | | | | | |
| **Age category (in years)** | | | | | | |
| 15–19 | 1 | | | 1 | | |
| 20–24 | 2.64* | 2.22 | 3.14 | 2.14* | 1.86 | 2.46 |
| 25–29 | 3.13* | 2.62 | 3.75 | 2.98* | 2.58 | 3.43 |
| 30–34 | 4.23* | 3.51 | 5.10 | 4.83* | 4.18 | 5.58 |
| 35–39 | 5.99* | 4.97 | 7.22 | 6.50* | 5.62 | 7.53 |
| 40–44 | 9.49* | 7.88 | 11.42 | 8.74* | 7.55 | 10.13 |
| 45–49 | 12.51* | 10.41 | 15.04 | 9.76* | 8.43 | 11.30 |
| 50–54 | 17.29* | 14.36 | 20.81 | 10.56* | 9.10 | 12.25 |
| **Marital status** | | | | | | |
| Never married | 1 | | | 1 | | |
| Currently married | 0.95 | 0.86 | 1.05 | 1.12* | 1.04 | 1.21 |
| Widowed/Separated/Divorced | 1.09 | 0.90 | 1.32 | 1.15 | 0.98 | 1.36 |
| **Religion** | | | | | | |
| Hindu | 1 | | | 1 | | |
| Muslim | 0.90* | 0.83 | 0.98 | 0.86* | 0.81 | 0.92 |
| Christian | 0.92 | 0.79 | 1.06 | 0.89* | 0.78 | 1.00 |
| Others* | 1.29* | 1.11 | 1.49 | 1.00 | 0.88 | 1.13 |
| **Educational status** | | | | | | |
| No education | 1 | | | 1 | | |
| Primary | 1.14* | 1.04 | 1.25 | 1.09* | 1.01 | 1.18 |
| Secondary | 1.20* | 1.11 | 1.30 | 1.24* | 1.16 | 1.33 |
| Higher | 1.01 | 0.91 | 1.12 | 1.12* | 1.03 | 1.21 |
| **Wealth Index of the household** | | | | | | |
| Poorest (I quintile) | 1 | | | 1 | | |
| Poorer (II quintile) | 1.01 | 0.93 | 1.11 | 1.17* | 1.08 | 1.26 |
| Middle (III quintile) | 1.07 | 0.97 | 1.18 | 1.32* | 1.22 | 1.43 |
| Richer (IV quintile) | 1.12* | 1.01 | 1.23 | 1.32* | 1.22 | 1.43 |
| Richest (V quintile) | 1.14* | 1.02 | 1.28 | 1.35* | 1.23 | 1.49 |
| **Current occupation** | | | | | | |
| Unemployed | 1 | | | 1 | | |
| Employed | 0.95 | 0.88 | 1.03 | 0.97 | 0.91 | 1.04 |
| **Residence** | | | | | | |
| Urban | 1 | | | 1 | | |
| Rural | 0.92* | 0.86 | 0.97 | 0.92* | 0.88 | 0.97 |
| **Region** | | | | | | |
| North | 1 | | | 1 | | |
| Central | 1.05 | 0.93 | 1.17 | 1.06 | 0.97 | 1.17 |
| East | 0.66* | 0.59 | 0.73 | 0.67* | 0.62 | 0.74 |
| Northeast | 1.01 | 0.88 | 1.16 | 0.97 | 0.86 | 1.09 |
| West | 0.75* | 0.68 | 0.83 | 0.95 | 0.88 | 1.04 |
| South | 1.03 | 0.93 | 1.13 | 1.27* | 1.17 | 1.38 |

*(Continued)*

**Table 2.** (Continued)

| Characteristics | SBP (>/=140 mmHg) | | | DBP (>/=90 mmHg) | | |
|---|---|---|---|---|---|---|
| | OR | 95% CI | | OR | 95% CI | |
| | | Lower | Upper | | Lower | Upper |
| **Caste** | | | | | | |
| SC | 1 | | | 1 | | |
| ST | 1.15* | 1.04 | 1.28 | 1.04 | 0.95 | 1.12 |
| OBC | 0.95 | 0.89 | 1.02 | 0.89* | 0.84 | 0.94 |
| Others | 1.01 | 0.93 | 1.09 | 0.91* | 0.85 | 0.97 |
| **Behavioral and physiological characteristics** | | | | | | |
| **Waist Hip Ratio (WHR) category** | | | | | | |
| <0.90 | 1 | | | 1 | | |
| >/=0.90 | 1.46* | 1.39 | 1.54 | 1.62* | 1.55 | 1.69 |
| **History of Alcohol use** | | | | | | |
| No | 1 | | | 1 | | |
| Yes | 1.28* | 1.20 | 1.35 | 1.50* | 1.43 | 1.57 |
| **History of tobacco use** | | | | | | |
| No | 1 | | | 1 | | |
| Yes | 1.20* | 1.11 | 1.29 | 1.05 | 0.98 | 1.11 |
| **Frequency of fruit intake** | | | | | | |
| Never | 1 | | | 1 | | |
| Daily | 1.15 | 0.93 | 1.42 | 0.80* | 0.68 | 0.95 |
| Weekly | 1.09 | 0.89 | 1.33 | 0.91 | 0.78 | 1.06 |
| Occasionally | 1.05 | 0.86 | 1.29 | 0.85* | 0.73 | 0.99 |
| **Frequency of fried food intake** | | | | | | |
| Never | 1 | | | 1 | | |
| Daily | 1.18* | 1.04 | 1.33 | 0.90* | 0.81 | 0.99 |
| Weekly | 0.99 | 0.89 | 1.09 | 0.91* | 0.84 | 0.98 |
| Occasionally | 1.08 | 0.99 | 1.19 | 0.93* | 0.86 | 1.00 |
| **Co-morbidities** | | | | | | |
| Diabetes | 1.12 | 1.00 | 1.25 | 1.01 | 0.92 | 1.12 |
| Cardiovascular disease | 0.81 | 0.65 | 1.00 | 0.84 | 0.70 | 1.01 |

*p value<0.05.

Table 7 presents the diagnostic performance of age as a predictor of hypertension using ROC analysis for both systolic and diastolic blood pressure in males and females. The analysis identifies 36.5 years as the optimal cut-off age for initiating hypertension screening in both sexes, based on the Youden Index. Among males, the area under the curve (AUC) was 0.71 for systolic BP and 0.69 for diastolic BP, indicating fair discriminatory ability. In females, the AUC values were slightly lower at 0.68 for systolic BP and 0.66 for diastolic BP. Sensitivity and specificity values at this cut-off were reasonably balanced, ranging from 56% to 61% sensitivity and 47% to 50% specificity. These findings support 35–36 years as a data-driven and practical threshold for routine hypertension screening among Indian adults.

## Discussion

The present study utilized data from the NFHS-5 survey, which includes data from 1,01,839 men and 7,24,115 women participants. Hypertension, identified in 28.5% of Indian adults aged 18–69, remains a major public health challenge,

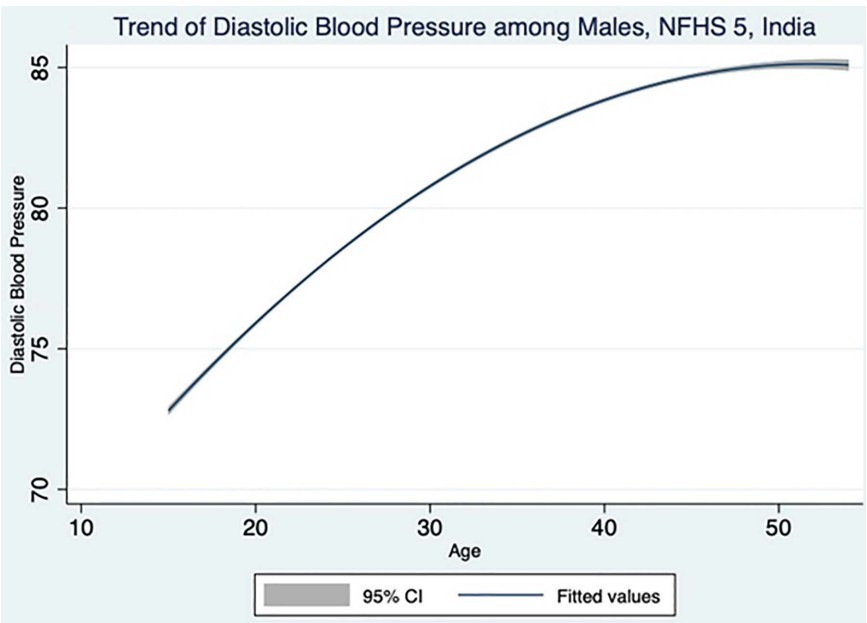

**Fig 2. Male diastolic blood pressure across different age group in the study population.**

contributing significantly to the rising burden of non-communicable diseases (NCDs) in India [1,2]. The findings emphasize the need for targeted screening interventions, particularly focusing on desired age for screening, along with socio-demographic, behavioral, and physiological factors.

## Comparison with global and national trends

Globally, the prevalence of hypertension among adults was 32% in women and 34% in men in 2019, with 41% of women and 51% of men being unaware of their hypertensive status [1]. This study's findings align with global trends, showing a linear increase in blood pressure with age, emphasizing the need for timely detection. The study identifies the age group of 35–39 years as a critical point for initiating hypertension screening, suggesting that approximately 60% of adults in this age group have elevated blood pressure, a statistic that parallels the 60% sensitivity identified in the current study [5].

The World Health Organization (WHO) emphasizes opportunistic screening for all adults in primary care settings but does not recommend a specific age threshold [18]. The US Preventive Services Task Force (USPSTF) recommends annual hypertension screening for adults aged ≥40 years or younger adults with risk factors, and every 3–5 years for those aged 18–39 years without risk factors [5]. The Indian NPCDCS program currently recommends annual screening starting at 30 years of age [19], but this recommendation lacks age-stratified evidence from large-scale national datasets. Our study contributes by empirically identifying 36.5 years as the optimal cut-off based on ROC analysis, offering the first age-specific evidence from a nationally representative Indian sample (NFHS-5). Compared to prior Indian studies such as Prenissl et al. (2019), which highlighted awareness gaps but did not define an ideal screening age [20], our findings offer direct policy guidance.

In India, the National Family Health Survey (NFHS-5) data highlights a gap in hypertension awareness, with only 27% of hypertensive individuals aware of their condition, 14.5% receiving treatment, and just 12.6% having controlled blood pressure [2]. These figures underscore the need for improved screening strategies. This study further suggests that start-ing screening at 35 years could potentially capture a significant portion of undiagnosed cases, contributing to the national

**Table 3.** Distribution of systolic and diastolic blood pressure among females according to socio-demographic, behavioral and physiological characteristics of the study population, NFHS-5, 2019−21, India (N = 7,24,115).

| Characteristics | N | (%) | Systolic blood pressure | | Diastolic blood pressure | |
|---|---|---|---|---|---|---|
| | | | Weighted mean | 95%CI | Weighted mean | 95%CI |
| **Socio-demography** | | | | | | |
| **Age category (in years)** | | | | | | |
| 15–19 | 111820 | 15.4 | 109.47 | 109.40-109.53 | 72.34 | 72.29-72.39 |
| 20–24 | 109961 | 15.2 | 110.66 | 110.59-110.72 | 73.99 | 73.94-74.04 |
| 25–29 | 108578 | 15.0 | 112.42 | 112.35-112.49 | 75.84 | 75.79-75.89 |
| 30–34 | 93505 | 12.9 | 115.09 | 115.01-115.17 | 77.93 | 77.87-77.98 |
| 35–39 | 90926 | 12.6 | 117.86 | 117.77-117.95 | 79.68 | 79.62-79.75 |
| 40–44 | 76627 | 10.6 | 121.43 | 121.32-121.53 | 81.25 | 81.18-81.32 |
| 45–49 | 79726 | 11.0 | 124.71 | 124.59-124.83 | 82.27 | 82.20-82.34 |
| **Marital status** | | | | | | |
| Never married | 155694 | 21.5 | 110.37 | 110.31-110.42 | 73.46 | 73.42-73.50 |
| Currently married | 486713 | 67.2 | 116.51 | 116.47-116.55 | 78.14 | 78.11-78.16 |
| Widowed/Separated/Divorced | 28736 | 4.0 | 120.02 | 119.83-120.21 | 80.42 | 80.30-80.54 |
| **Religion** | | | | | | |
| Hindu | 550204 | 76.0 | 115.09 | 115.05-115.13 | 77.08 | 77.06-77.11 |
| Muslim | 86887 | 12.0 | 115.80 | 115.71-115.90 | 77.28 | 77.21-77.34 |
| Christian | 15773 | 2.2 | 114.55 | 114.31-114.79 | 77.26 | 77.09-77.41 |
| Others* | 18280 | 2.5 | 117.60 | 117.39-117.82 | 78.46 | 78.32-78.61 |
| **Educational status** | | | | | | |
| No education | 152941 | 21.1 | 118.82 | 118.74-118.89 | 79.32 | 79.27-79.37 |
| Primary | 79627 | 11.0 | 117.47 | 117.37-117.57 | 78.68 | 78.62-78.75 |
| Secondary | 336361 | 46.5 | 113.84 | 113.79-113.88 | 76.18 | 76.14-76.21 |
| Higher | 102214 | 14.1 | 112.74 | 112.66-112.82 | 75.91 | 75.85-75.96 |
| **Wealth Index of the household** | | | | | | |
| Poorest (I quintile) | 125420 | 17.3 | 115.07 | 114.99-115.14 | 76.55 | 76.50-76.61 |
| Poorer (II quintile) | 135405 | 18.7 | 114.83 | 114.76-114.91 | 76.76 | 76.71-76.81 |
| Middle (III quintile) | 139011 | 19.2 | 114.99 | 114.92-115.06 | 77.15 | 77.10-77.20 |
| Richer (IV quintile) | 140220 | 19.4 | 115.30 | 115.22-115.37 | 77.45 | 77.40-77.50 |
| Richest (V quintile) | 131087 | 18.1 | 116.01 | 115.94-116.09 | 77.80 | 77.75-77.85 |
| **Current occupation** | | | | | | |
| Unemployed | 74844 | 10.3 | 114.77 | 114.67-114.87 | 76.82 | 76.75-76.89 |
| Employed | 25627 | 3.5 | 116.20 | 116.02-116.37 | 78.11 | 77.99-78.22 |
| **Residence** | | | | | | |
| Urban | 212706 | 29.4 | 115.44 | 115.38-115.50 | 77.52 | 77.48-77.56 |
| Rural | 458437 | 63.3 | 115.14 | 115.10-115.18 | 76.97 | 76.95-77.00 |
| **Region** | | | | | | |
| North | 95548 | 13.2 | 117.40 | 117.31-117.48 | 78.04 | 77.98-78.09 |
| Central | 168057 | 23.2 | 116.32 | 116.25-116.38 | 77.54 | 77.49-77.58 |
| East | 152178 | 21.0 | 114.33 | 114.26-114.40 | 76.40 | 76.36-76.45 |
| Northeast | 24576 | 3.4 | 116.89 | 116.71-117.07 | 77.62 | 77.50-77.74 |
| West | 93541 | 12.9 | 114.44 | 114.35-114.53 | 76.88 | 76.82-76.94 |
| South | 137243 | 19.0 | 113.65 | 113.58-113.73 | 76.98 | 76.93-77.03 |
| **Caste** | | | | | | |
| SC | 148158 | 20.5 | 114.53 | 114.46-114.60 | 76.87 | 76.82-76.92 |

*(Continued)*

**Table 3.** (Continued)

| Characteristics | N | (%) | Systolic blood pressure | | Diastolic blood pressure | |
|---|---|---|---|---|---|---|
| | | | Weighted mean | 95%CI | Weighted mean | 95%CI |
| ST | 63553 | 8.8 | 116.57 | 116.47-116.68 | 77.57 | 77.50-77.65 |
| OBC | 290700 | 40.1 | 114.94 | 114.89-114.99 | 76.98 | 76.95-77.02 |
| Others | 168734 | 23.3 | 115.86 | 115.80-115.93 | 77.52 | 77.47-77.56 |
| **Behavioral and physiological** | | | | | | |
| **Waist Hip Ratio category** | | | | | | |
| >/=0.85 | 382488 | 52.8 | 116.87 | 116.82-116.91 | 78.06 | 78.03-78.09 |
| <0.85 | 257040 | 35.5 | 112.80 | 112.75-112.85 | 75.78 | 75.75-75.82 |
| **History of Alcohol use** | | | | | | |
| No | 666052 | 92.0 | 115.20 | 115.17-115.24 | 77.12 | 77.10-77.15 |
| Yes | 5092 | 0.7 | 119.76 | 119.32-120.21 | 80.38 | 80.09-80.68 |
| **History of tobacco use** | | | | | | |
| No | 670500 | 92.6 | 115.24 | 115.20-115.27 | 77.15 | 77.12-77.17 |
| Yes | 644 | 0.1 | 115.62 | 114.43-116.80 | 77.57 | 76.83-78.31 |
| **Frequency of fruit intake** | | | | | | |
| Never | 10901 | 1.5 | 115.49 | 115.22-115.77 | 77.44 | 77.25-77.62 |
| Daily | 81322 | 11.2 | 114.68 | 114.59-114.78 | 76.95 | 76.89-77.02 |
| Weekly | 247699 | 34.2 | 115.23 | 115.17-115.28 | 77.20 | 77.16-77.24 |
| Occasionally | 331221 | 45.7 | 115.37 | 115.32-115.42 | 77.15 | 77.12-77.18 |
| **Frequency of fried food intake** | | | | | | |
| Never | 29516 | 4.1 | 116.03 | 115.86-116.20 | 77.79 | 77.67-77.90 |
| Daily | 49013 | 6.8 | 115.10 | 114.97-115.23 | 77.05 | 76.96-77.13 |
| Weekly | 237557 | 32.8 | 115.13 | 115.07-115.19 | 77.05 | 77.01-77.09 |
| Occasionally | 355057 | 49.0 | 115.26 | 115.22-115.31 | 77.17 | 77.14-77.21 |
| **Co-morbidities** | | | | | | |
| Hypertension | 32171 | 4.4 | 124.18 | 123.97-124.40 | 82.96 | 82.83-83.10 |
| Diabetes | 12725 | 1.8 | 123.84 | 123.54-124.14 | 82.51 | 82.34-82.69 |
| Cardiovascular disease | 4871 | 0.7 | 117.99 | 117.54-118.43 | 79.24 | 78.95-79.53 |

*Includes Sikh, Buddhist, Jain, Jewish and others.

goal of reducing hypertension prevalence by 25% by 2025 [21].This gap in awareness and management is concerning, especially considering the high prevalence of hypertension, which is a major risk factor for cardiovascular diseases, including stroke and heart attacks [22].

One of the critical challenges in managing hypertension in India is the low rate of awareness. Studies have shown that lack of awareness is often due to the asymptomatic nature of hypertension, where individuals do not experience symptoms until significant damage has occurred to vital organs [23]. This underlines the importance of regular screening to identify individuals with elevated blood pressure before complications arise.

Moreover, the treatment rates for hypertension are alarmingly low. Even among those diagnosed, the rate of treatment is just 14.5%, indicating significant barriers to accessing healthcare, such as economic constraints, lack of healthcare infrastructure, and inadequate follow-up [24]. Cultural beliefs and practices also play a role in influencing health-seeking behavior, with many individuals relying on traditional medicine or home remedies instead of seeking formal medical treatment [25].

**Table 4. Multivariate logistic regression analysis to determine the factors associated with elevated systolic and diastolic blood pressure among female, NFHS-5, 2019–21, India.**

| Characteristics | SBP (>/=140 mmHg) | | | DBP (>/=90 mmHg) | | |
|---|---|---|---|---|---|---|
| | OR | 95% CI | | OR | 95% CI | |
| | | Lower | Upper | | Lower | Upper |
| **Socio-demographic characteristics** | | | | | | |
| **Age category (in years)** | | | | | | |
| 15–19 | 1 | | | 1 | | |
| 20–24 | 1.88* | 1.40 | 2.52 | 1.71* | 1.45 | 2.01 |
| 25–29 | 3.89* | 2.91 | 5.21 | 2.49* | 2.10 | 2.95 |
| 30–34 | 7.25* | 5.43 | 9.68 | 4.10* | 3.46 | 4.86 |
| 35–39 | 12.76* | 9.60 | 16.96 | 6.77* | 5.73 | 8.00 |
| 40–44 | 24.53* | 18.48 | 32.55 | 9.01* | 7.61 | 10.66 |
| 45–49 | 37.07* | 27.97 | 49.13 | 11.16* | 9.44 | 13.20 |
| **Marital status** | | | | | | |
| Never married | 1 | | | 1 | | |
| Currently married | 0.79* | 0.66 | 0.95 | 1.05 | 0.93 | 1.19 |
| Widowed/Separated/Divorced | 0.95 | 0.77 | 1.18 | 1.12 | 0.96 | 1.30 |
| **Religion** | | | | | | |
| Hindu | 1 | | | 1 | | |
| Muslim | 1.13* | 1.03 | 1.25 | 1.16* | 1.08 | 1.25 |
| Christian | 0.90 | 0.74 | 1.11 | 0.97 | 0.84 | 1.13 |
| Others | 1.45* | 1.22 | 1.71 | 1.32* | 1.16 | 1.51 |
| **Educational status** | | | | | | |
| No education | 1 | | | 1 | | |
| Primary | 0.91 | 0.83 | 1.01 | 1.07 | 0.99 | 1.15 |
| Secondary | 0.92* | 0.84 | 1.00 | 0.99 | 0.93 | 1.06 |
| Higher | 0.72* | 0.62 | 0.82 | .80* | 0.72 | 0.88 |
| **Wealth Index of the household** | | | | | | |
| Poorest (I quintile) | 1 | | | 1 | | |
| Poorer (II quintile) | 0.96 | 0.86 | 1.06 | 1.08 | 0.99 | 1.17 |
| Middle (III quintile) | 1.07 | 0.96 | 1.20 | 1.13* | 1.04 | 1.23 |
| Richer (IV quintile) | 1.13 | 1.00 | 1.27 | 1.25* | 1.14 | 1.37 |
| Richest (V quintile) | 1.12 | 0.98 | 1.29 | 1.23* | 1.11 | 1.37 |
| **Current occupation** | | | | | | |
| Unemployed | 1 | | | 1 | | |
| Employed | 0.86* | 0.80 | 0.92 | 0.95 | 0.90 | 1.00 |
| **Residence** | | | | | | |
| Urban | 1 | | | 1 | | |
| Rural | 0.98 | 0.90 | 1.06 | 0.95 | 0.90 | 1.01 |
| **Region** | | | | | | |
| North | 1 | | | 1 | | |
| Central | 1.29* | 1.15 | 1.44 | 1.22* | 1.12 | 1.33 |
| East | 1.01 | 0.90 | 1.14 | 0.97 | 0.88 | 1.06 |
| Northeast | 1.28* | 1.07 | 1.54 | 1.17* | 1.01 | 1.35 |
| West | .94 | 0.83 | 1.07 | 1.07 | 0.98 | 1.17 |
| South | 1.20* | 1.07 | 1.34 | 1.32* | 1.21 | 1.43 |

*(Continued)*

**Table 4.** (Continued)

| Characteristics | SBP (>/=140 mmHg) | | | DBP (>/=90 mmHg) | | |
|---|---|---|---|---|---|---|
| | OR | 95% CI | | OR | 95% CI | |
| | | Lower | Upper | | Lower | Upper |
| **Caste** | | | | | | |
| SC | 1 | | | 1 | | |
| ST | 1.27* | 1.12 | 1.44 | 1.16* | 1.06 | 1.28 |
| OBC | 0.96 | 0.88 | 1.04 | 0.92* | 0.86 | 0.98 |
| Others | 1.10 | 1.00 | 1.22 | 1.03 | 0.95 | 1.10 |
| **Behavioral and physiological characteristics** | | | | | | |
| **Waist Hip Ratio (WHR) category** | | | | | | |
| <0.85 | 1 | | | 1 | | |
| >/=0.85 | 1.38* | 1.28 | 1.48 | 1.38* | 1.31 | 1.46 |
| **History of Alcohol use** | | | | | | |
| No | 1 | | | 1 | | |
| Yes | 1.44* | 1.13 | 1.84 | 1.56* | 1.28 | 1.89 |
| **History of tobacco use** | | | | | | |
| No | 1 | | | 1 | | |
| Yes | 1.01 | 0.38 | 2.73 | 0.44 | 0.15 | 1.27 |
| **Frequency of fruit intake** | | | | | | |
| Never | 1 | | | 1 | | |
| Daily | 0.96 | 0.75 | 1.23 | 0.83 | 0.69 | 1.00 |
| Weekly | 1.00 | 0.79 | 1.26 | 0.88 | 0.74 | 1.05 |
| Occasionally | 0.93 | 0.74 | 1.18 | 0.85 | 0.72 | 1.02 |
| **Frequency of fried food intake** | | | | | | |
| Never | 1 | | | 1 | | |
| Daily | 0.85 | 0.71 | 1.01 | 0.87* | 0.75 | 0.99 |
| Weekly | 0.76* | 0.66 | 0.88 | 0.83* | 0.75 | 0.93 |
| Occasionally | 0.81* | 0.71 | 0.93 | 0.86* | 0.77 | 0.96 |
| **Co-morbidities** | | | | | | |
| Diabetes | 1.53* | 1.32 | 1.77 | 1.60* | 1.42 | 1.81 |
| Cardiovascular disease | 1.06 | 0.80 | 1.41 | 1.00 | 0.79 | 1.25 |

*p value<0.05.

The control rate, with only 12.6% of hypertensive individuals maintaining their blood pressure within recommended levels, highlights the challenges in ongoing management and adherence to treatment regimens [2]. Poor adherence to prescribed medication, often due to side effects, lack of understanding of the chronic nature of hypertension, and the financial burden of long-term treatment, are significant factors contributing to this issue [26].

Given these challenges, the study suggests that initiating hypertension screening at age 35 could play a crucial role in capturing a significant portion of undiagnosed cases. The age group of 35–39 years is identified as a critical period where the prevalence of hypertension begins to rise sharply. Within this age group, the study found that 56.4% of males and 60.9% of females have elevated blood pressure, underscoring the importance of early detection [27].Early screening is essential because evidence shows that hypertension detected and managed early significantly reduces the risk of cardiovascular events later in life. For example, the Framingham Heart Study demonstrated that individuals who developed

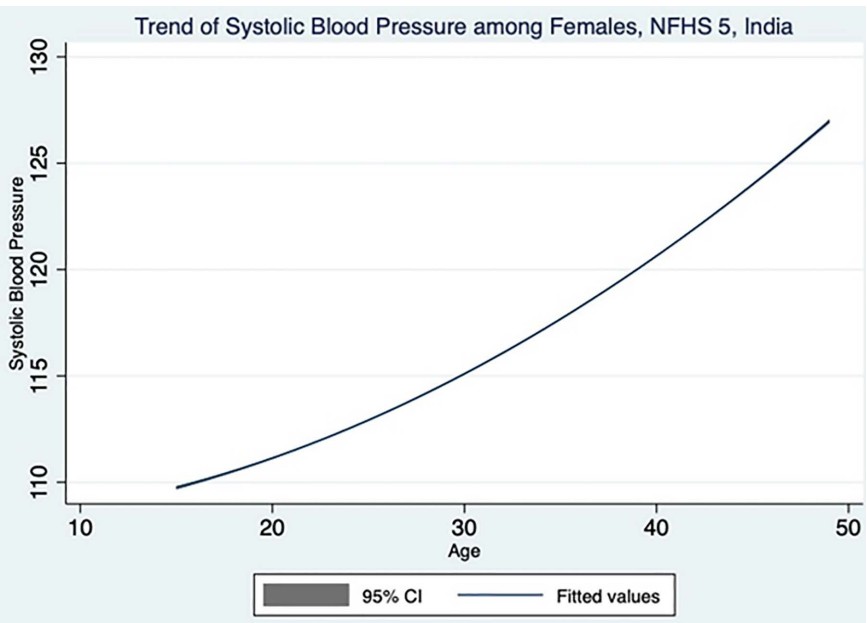

**Fig 3. Female systolic blood pressure across different age group in the study population.**

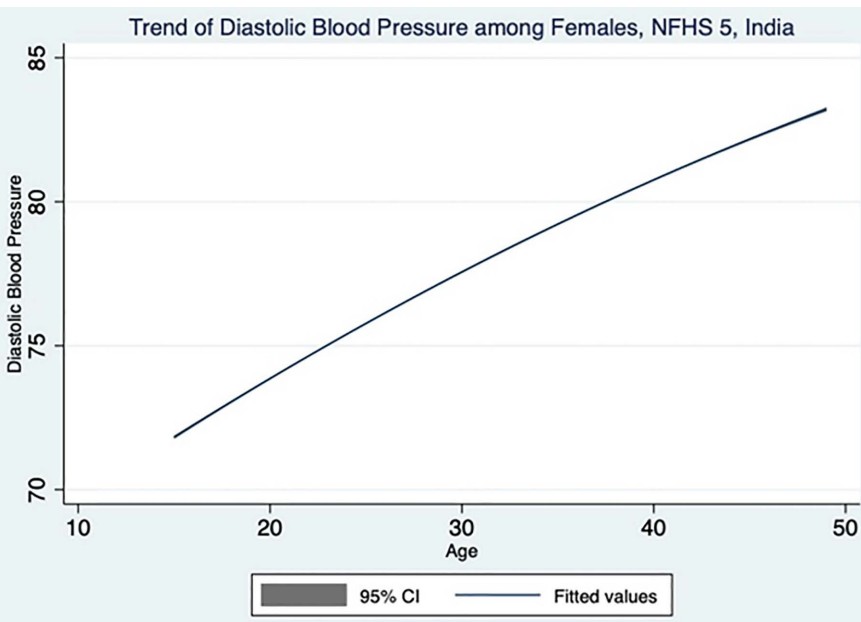

**Fig 4. Female diastolic blood pressure across different age group in the study population.**

**Table 5. Cut-off in years for males based on ROC curves.**

| Age category (in years) | SBP (>/=140 mmHg) | | | DBP (>/=90 mmHg) | | |
|---|---|---|---|---|---|---|
| | Cut off | Sensitivity | Specificity | Cut off | Sensitivity | Specificity |
| 15–19 | 16.5 | 70.0% | 41.8% | 17.5 | 52.6% | 61.4% |
| 20–24 | 21.5 | 63.2% | 41.1% | 21.5 | 66.5% | 41.3% |
| 25–29 | 26.5 | 60.1% | 43.2% | 26.5 | 61.2% | 43.5% |
| 30–34 | 31.5 | 60.6% | 44.4% | 31.5 | 57.9% | 44.4% |
| 35–39 | 36.5 | 57.9% | 49.7% | 36.5 | 56.5% | 50.2% |
| 40–44 | 41.5 | 57.0% | 48.2% | 40.5 | 70.3% | 33.3% |
| 45–49 | 46.5 | 57.1% | 49.5% | 46.5 | 55.0% | 49.4% |
| 50–54 | 51.5 | 58.1% | 44.5% | 50.5 | 74.6% | 27.1% |
| 15–54 (overall) | 35.5 | 68.0% | 63.9% | 30.5 | 78.8% | 52.3% |

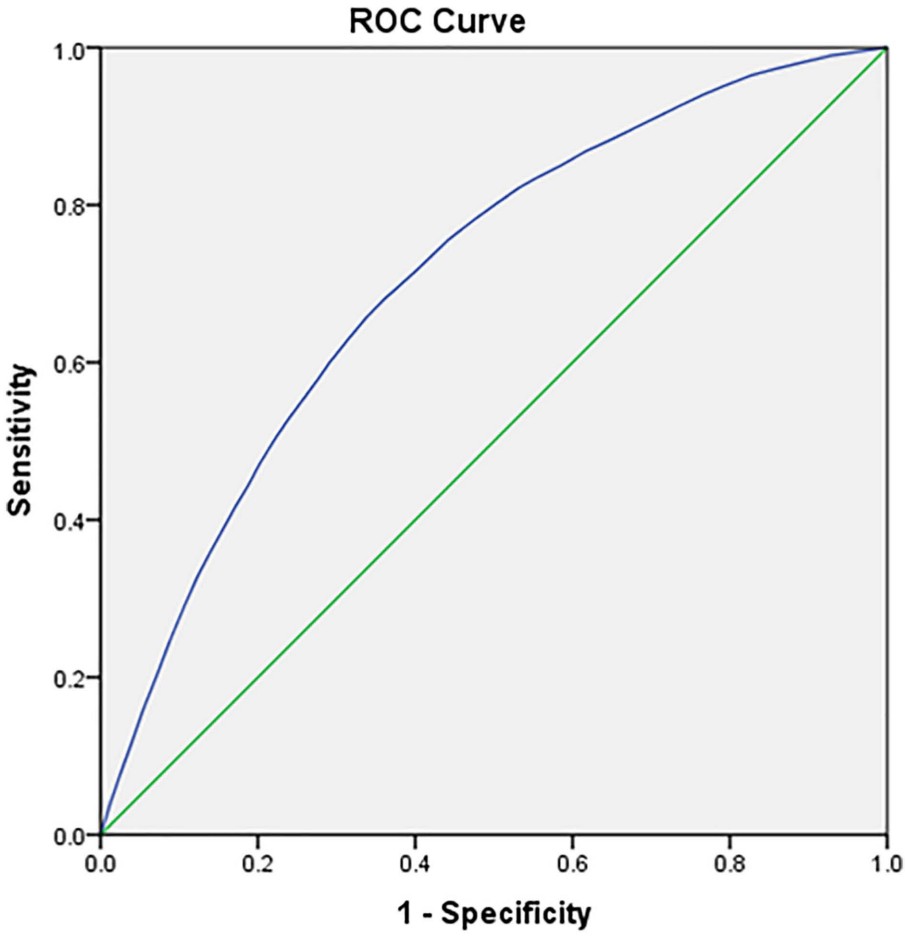

**Fig 5. Male systolic blood pressure across different age group in the study population.**

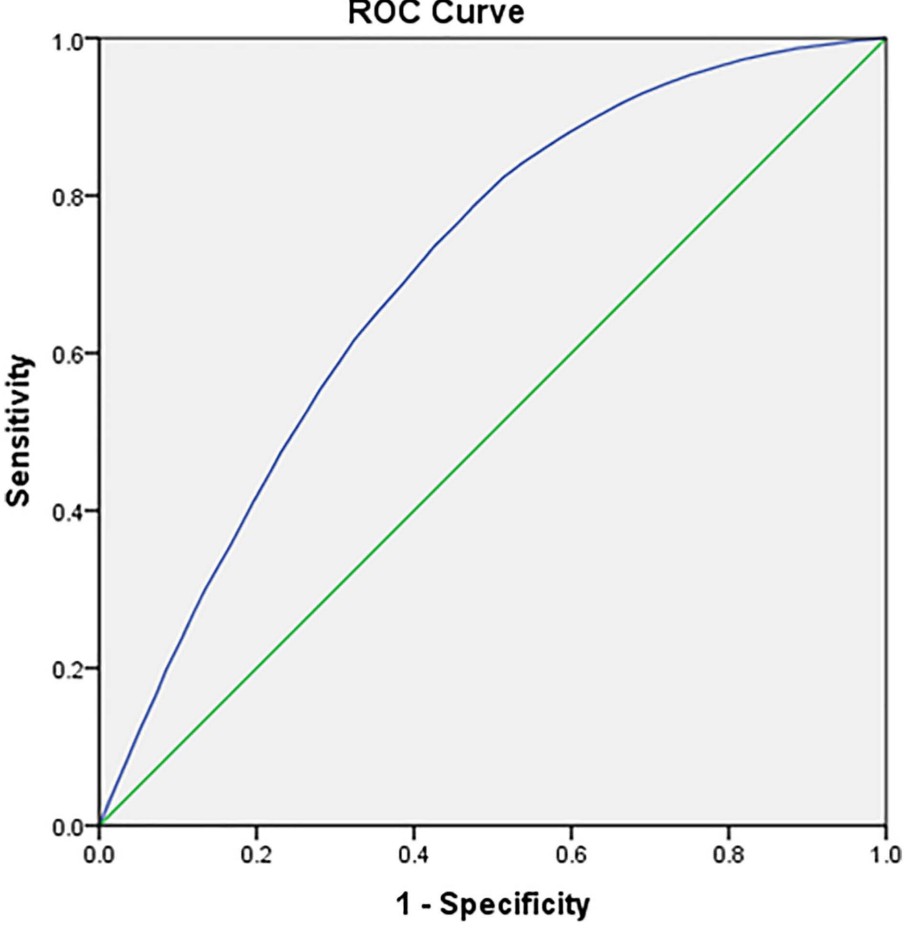

## ROC Curve

*Diagonal segments are produced by ties.*

**Fig 6. Male diastolic blood pressure across different age group in the study population.**

**Table 6. Cut-off in years for females based on ROC curves.**

| Age category (in years) | SBP (>/=140 mmHg) | | | DBP (>/=90 mmHg) | | |
|---|---|---|---|---|---|---|
| | Cut off | Sensitivity | Specificity | Cut off | Sensitivity | Specificity |
| 15–19 | 16.5 | 66.3% | 40.4% | 16.5 | 64.2% | 40.4% |
| 20–24 | 21.5 | 64.1% | 40.4% | 21.5 | 65.4% | 40.6% |
| 25–29 | 26.5 | 63.1% | 43.7% | 26.5 | 62.6% | 43.9% |
| 30–34 | 31.5 | 63.1% | 44.5% | 31.5 | 61.1% | 44.7% |
| 35–39 | 36.5 | 60.6% | 47.4% | 36.5 | 57.8% | 47.6% |
| 40–44 | 41.5 | 59.4% | 46.2% | 41.5 | 57.5% | 46.2% |
| 45–49 | 46.5 | 59.3% | 46.4% | 46.5 | 56.9% | 46.1% |
| 15–49 (overall) | 34.5 | 78.2% | 65.4% | 32.5 | 72.5% | 61.5% |

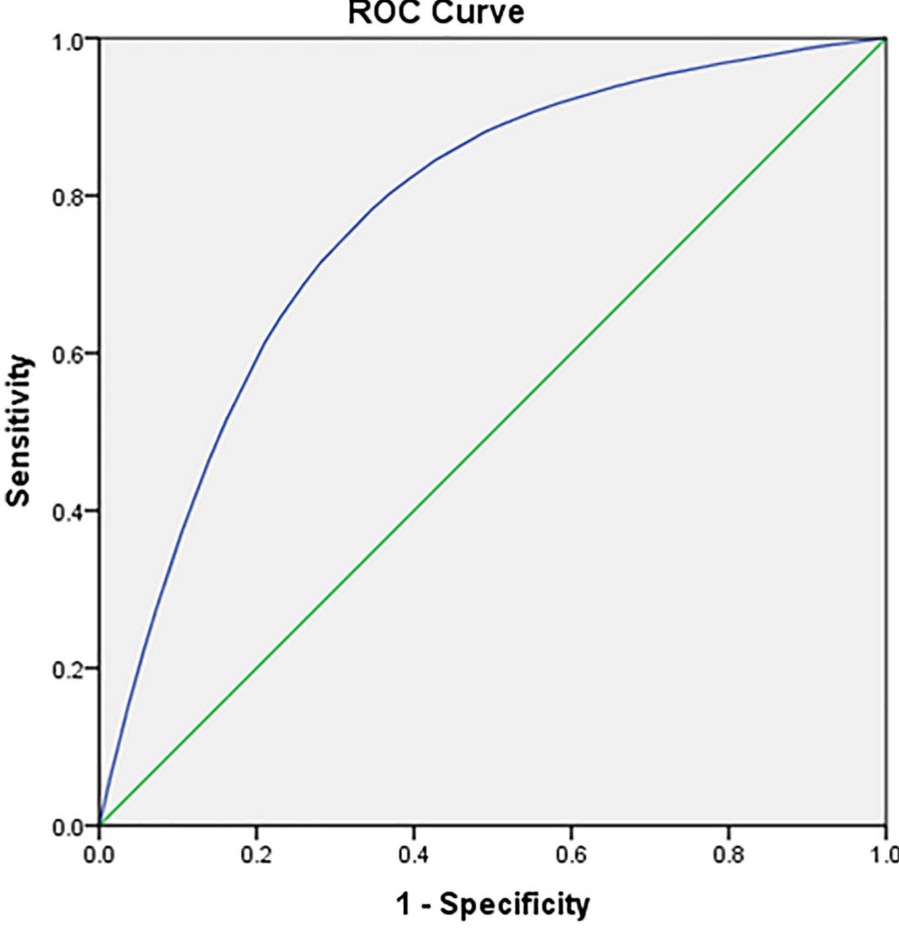

**Fig 7. Female systolic blood pressure across different age group in the study population.**

hypertension by their mid-30s had a markedly higher lifetime risk of cardiovascular disease, including heart failure, myocardial infarction, and stroke [28].

The benefits of early screening are further supported by findings from the SPRINT trial, which showed that intensive blood pressure management in individuals at risk significantly reduced the rates of cardiovascular events and all-cause mortality [29]. Additionally, a systematic review of hypertension management strategies in low- and middle-income countries highlighted that early detection through community-based screening programs effectively reduces the burden of hypertension-related complications [30].

### Socio-demographic determinants

This study reveals a significant association between marital status and diastolic blood pressure (DBP), with married men exhibiting a 12% increased risk of elevated DBP compared to their unmarried counterparts. This finding resonates with previous studies indicating that married individuals, particularly men, may experience higher blood pressure due to marital

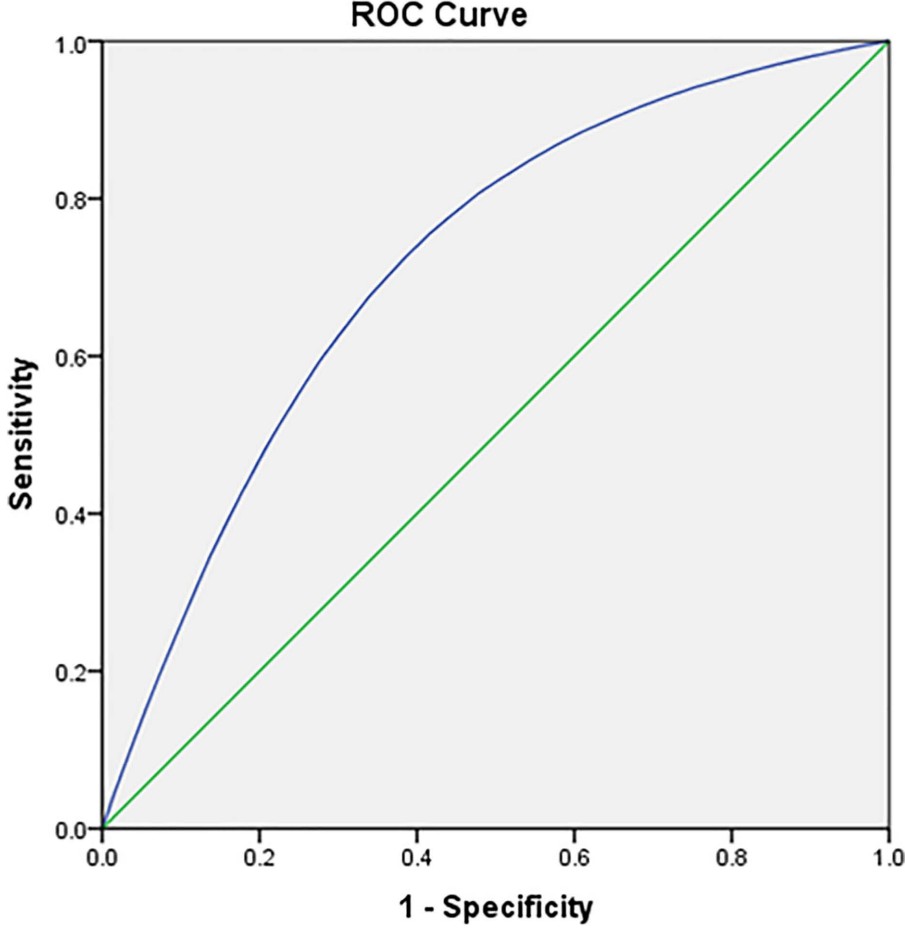

### ROC Curve

**Fig 8. Female diastolic blood pressure across different age group in the study population.**

**Table 7. ROC-Based Diagnostic Performance of Age for Predicting Hypertension in Indian Adults (NFHS-5, 2019–21).**

| Sex | Outcome | Cut-off Age (years) | AUC (95% CI) | Sensitivity (%) | Specificity (%) | Youden Index |
|---|---|---|---|---|---|---|
| **Male** | SBP ≥ 140 mmHg | 36.5 | 0.71 (0.70–0.72) | 57.9 | 49.7 | 0.076 |
| | DBP ≥ 90 mmHg | 36.5 | 0.69 (0.68–0.70) | 56.5 | 50.2 | 0.067 |
| **Female** | SBP ≥ 140 mmHg | 36.5 | 0.68 (0.67–0.69) | 60.6 | 47.4 | 0.08 |
| | DBP ≥ 90 mmHg | 36.5 | 0.66 (0.65–0.67) | 57.8 | 47.6 | 0.054 |

stress [31]. However, this stress-related increase in DBP needs to be balanced against the generally recognized health benefits of marriage, such as better social support and health behaviors [32].

The study also highlights a 10% lower risk of hypertension among Muslims compared to Hindus. This difference could be attributed to religious practices such as fasting during Ramadan, which has been shown to have beneficial effects on cardiovascular health [27]. Further, dietary habits prevalent in Muslim communities, such as lower consumption of alcohol and possibly higher consumption of whole grains and legumes, may contribute to this reduced risk [33].

Education and wealth index were also significant predictors of hypertension. The study found that individuals with no formal education had a 20% higher risk of hypertension compared to those with higher education. This is consistent with global data suggesting that lower educational attainment is associated with poorer health outcomes due to factors like reduced health literacy and access to healthcare services [34]. Moreover, the wealth index showed a clear gradient, with individuals in the highest quintile being 35% more likely to have elevated DBP, possibly due to lifestyle factors such as higher dietary intake of fats and sugars [35].

## Behavioral and physiological risk factors

Behavioral risk factors, particularly alcohol consumption and obesity, showed strong associations with hypertension in this study. Men who consumed alcohol were 28% more likely to have elevated systolic blood pressure (SBP), while those with a high waist-hip ratio (WHR) had a 46% higher risk of elevated SBP and a 62% higher risk of elevated DBP [36]. These findings are supported by global statistics that link excessive alcohol consumption to an increased risk of hypertension due to its effects on the cardiovascular system [12].

Obesity, as indicated by WHR, is a well-established risk factor for hypertension, with studies showing that a $5\,kg/m^2$ increase in body mass index (BMI) can increase the risk of hypertension by 49% [37]. The study's findings that higher WHR correlates with a substantial increase in hypertension risk emphasize the need for weight management interventions as part of hypertension prevention strategies.

Interestingly, the study reports a protective effect of fried food consumption on DBP, particularly in women, with those consuming fried foods weekly having a 24% reduced risk of elevated SBP and a 17% reduced risk of elevated DBP. This finding is contrary to the general understanding that fried foods, typically high in trans fats and sodium, contribute to hypertension [38]. This unexpected result suggests that other factors, such as the type of oils used or the presence of other healthy behaviors, might be influencing these outcomes. For instance, some traditional Indian cooking practices involve the use of oils rich in unsaturated fats, which might offset the negative effects of frying [39]. Further research is needed to disentangle these associations and understand the complex dietary patterns at play.

The study also highlights a gender difference in the impact of fruit intake on blood pressure, with daily fruit consumption associated with a 20% reduced risk of elevated DBP in men, but not in women. This difference could be due to variations in the types of fruits consumed or the overall dietary context in which fruit is eaten. Fruits are rich in potassium, which has been shown to counteract the effects of sodium and lower blood pressure [40]. However, gender differences in nutrient metabolism or hormone levels may modify this protective effect, warranting further investigation.

## Implications for public health and policy

In India, the Ministry of Health and Family Welfare (MoHFW) recommends hypertension screening starting at age 30 [21]. However, this study suggests that screening should be intensified from age 35 to capture those at risk earlier, as evidence suggests that undiagnosed hypertension in this group leads to increased cardiovascular morbidity later in life [31]. Studies have shown that hypertension in individuals aged 35–44 is associated with a significantly higher risk of cardiovascular events, including heart attacks and strokes, compared to those diagnosed later in life [27,32]. Furthermore, data from the Framingham Heart Study indicate that individuals who develop hypertension by their mid-30s have a higher lifetime risk of cardiovascular disease, emphasizing the importance of early detection and management [33]. Additional research from the INTERHEART study confirms that the early onset of hypertension, particularly before age 40, substantially increases the risk of myocardial infarction, making early and targeted screening interventions crucial [34].

Based on the identified ROC thresholds and age-specific risk profiles, we recommend revising India's hypertension screening guidelines to initiate routine screening from 35 years onward for both men and women. This adjustment balances feasibility with early detection impact, especially considering that SBP begins to rise substantially from this age group onward.

The study's findings also highlight the need for integrated public health strategies that address the underlying socio-economic and behavioral determinants of hypertension. For instance, public health campaigns should promote life-style changes such as reducing alcohol intake and managing body weight, particularly among younger adults. Given that only 14.5% of hypertensive individuals in India are receiving treatment, there is a clear need for improving access to care and ensuring that screening programs are effectively linked to treatment and follow-up services [2]. Moreover, educational interventions aimed at improving health literacy could play a crucial role in reducing hypertension prevalence. The study's findings that lower educational attainment and wealth are associated with higher hypertension risk underscore the importance of addressing these social determinants of health through targeted public health interventions [41].

### Limitations and future research

While this study provides valuable insights, it has limitations that should be acknowledged. The cross-sectional design limits the ability to establish causality between the identified risk factors and hypertension. Future research should focus on longitudinal studies to better understand the temporal relationships and causality [14]. Additionally, the reliance on self-reported data for alcohol and tobacco use may introduce recall bias, potentially leading to underestimation of these behaviors' impact on hypertension [42]. We acknowledge the possibility of regional disparities in healthcare access, awareness, and hypertension screening practices across different states and zones in India, which may influence screening rates and outcomes. This warrants further subnational analysis.

The study also did not include detailed dietary assessments, particularly sodium intake, which is a key contributor to hypertension. The average sodium intake in India is estimated to be over 10 grams per day, which is more than double the World Health Organization's recommended limit of 5 grams per day [43]. Future studies should incorporate detailed dietary data, including sodium and potassium intake, to better understand their roles in hypertension in the Indian context [44].

### Conclusion

This study, based on nationally representative NFHS-5 data, provides robust evidence supporting the initiation of routine hypertension screening at 35 years of age for both men and women in India. Using ROC analysis, we identified 36.5 years as the optimal age cutoff with balanced sensitivity and specificity, suggesting a practical and data-driven revision to current national screening guidelines. While the cross-sectional nature of the dataset limits causal inferences, our findings warrant validation through longitudinal studies to confirm the predictive value of this threshold over time. To improve screening uptake among younger adults—especially in underserved regions—targeted community awareness campaigns, workplace-based screening programs, and mobile health interventions integrated with frontline health worker outreach (e.g., ASHAs) should be prioritized. These strategies can facilitate earlier diagnosis, improve blood pressure control, and reduce the burden of cardiovascular disease across the adult population.

### Author contributions

**Conceptualization:** Prakash Mathiyalagen, Anand Rajagopal, Sonu Goel.

**Data curation:** Prakash Mathiyalagen, Anand Rajagopal, Kavita Vasudevan.

**Formal analysis:** Prakash Mathiyalagen, Anand Rajagopal, Kavita Vasudevan.

**Methodology:** Sridevi Gnanasekaran, Sonu Goel.

**Project administration:** Jayanta Bora.

**Software:** Sridevi Gnanasekaran.

**Supervision:** Prakash Mathiyalagen, Kavita Vasudevan, Sharath Burugina Nagaraja, Jayanta Bora, Sonu Goel.

**Validation:** Jayanta Bora.

**Visualization:** Jayanta Bora.

**Writing – original draft:** Sharath Burugina Nagaraja, Jayanta Bora.

**Writing – review & editing:** Sridevi Gnanasekaran.

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
