## [Decision Letter · Decision Letter 0]

13 May 2025

Dear Dr. Goel,

We look forward to receiving your revised manuscript.

Kind regards,

Preeti Kanawjia, MD

Academic Editor

PLOS ONE

Journal Requirements:

“None”

Reviewers' comments:

Reviewer's Responses to Questions

**Comments to the Author**

1. Is the manuscript technically sound, and do the data support the conclusions?

Reviewer #1: Yes

Reviewer #2: No

2. Has the statistical analysis been performed appropriately and rigorously?

Reviewer #1: Yes

Reviewer #2: No

3. Have the authors made all data underlying the findings in their manuscript fully available?

Reviewer #1: Yes

Reviewer #2: Yes

4. Is the manuscript presented in an intelligible fashion and written in standard English?

Reviewer #1: Yes

Reviewer #2: No

Reviewer #1: Overall Evaluation:

The manuscript presents a valuable study using NFHS-5 data to identify the optimal age for initiating hypertension screening in Indian adults. The findings are highly relevant for public health policymakers and contribute to the discourse on early detection of hypertension. The study employs robust statistical methods, including multivariate logistic regression and ROC analysis, to determine the recommended screening age of 35 years.

While the manuscript is methodologically sound, revisions are necessary to improve clarity, statistical reporting, and alignment with journal guidelines. Below are detailed comments and recommendations to enhance the manuscript.

Reviewer #2: The study still need major revisions in statistical parts the points are give below.•Title: Hypertension Screening across Different Age Groups in Indian Adults: Evidence from Nationally Representative Cross-Sectional Data

•Title must be as: An association study of individual level risk factors and Hypertension among the Indians: Evidence from Nationally Representative Cross-Sectional Data

•What is the rational of selecting the particular age group mentioned in the study?

•How does linearity prevail if the age is categorized into different group? If there is any diagrammatic or linear association statistics are applied?

• In or to get the particular confounding variable like age as a key factor one should conduct Cochran-Mantel-Haenszel test.

•If it is not possible then age should be included as independent factor in quadratic form.

•Inclusion of independent variables should be assessed through collinearity test. The model diagnostic should be evaluated with R²and Chi square log likelihood values with step wise improvement.

•Write few sentences about SBP and DBP in introduction section.

•Clearly state about the gap exist in your research study from earlier researches.

•If only 35-39 years is vulnerable and it is treated as gap. Then why the interaction part is missing in combination with other factors.

•If sleep quality and stress are not matters. If so then mediation and moderation effect must be apprised. Because SBP and DBP oscillates according to this factors.

•In behavioral part the physical exercise is missing

•It will be appropriate to conduct decomposition analysis to find out the GAP as the study has focused on the male and female framed in separate tables.

•Youden index, a statistic used in evaluating diagnostic test accuracy, specifically to find the optimal cut-off point for a continuous variable when classifying individuals into two groups. In this study the age is grouped as polytomous vaiables.

•Classification table would be appropriate in such case.

•The Youden index (J) is calculated as: J = Sensitivity + Specificity - 1. The value in specific position is not marked in ROC graph.

**Do you want your identity to be public for this peer review?** For information about this choice, including consent withdrawal, please see our Privacy Policy

Reviewer #1: **Yes: ** Dr. Nidhi Jaswal

Reviewer #2: No

---

## [Author Response · Author response to Decision Letter 1]

29 May 2025

Manuscript Title:

Hypertension Screening across Different Age Groups in Indian Adults: Evidence from Nationally Representative Cross-Sectional Data

Major Revisions Required

1. Introduction

• The introduction should clearly highlight the research gap in India’s current hypertension screening policies and how this study aims to fill that gap.

• The authors should briefly discuss existing global guidelines on age-based hypertension screening and whether similar studies have been conducted in other LMICs (low- and middle-income countries).

• The rationale behind selecting 35 years as the potential threshold should be better justified in light of previous research.

Reviewer Comment 1:

Author Response:

We thank the reviewer for this insightful suggestion. In response, we have substantially revised the Introduction section to better highlight the policy and evidence gap in India’s current hypertension screening practices. Although the National Programme for Prevention and Control of Cancer, Diabetes, Cardiovascular Diseases and Stroke (NPCDCS) recommends annual hypertension screening for adults aged 30 years and above [1], this age threshold has not been empirically validated using stratified national data. Our study directly addresses this gap by analyzing data from NFHS-5, a nationally representative dataset, to determine an evidence-based optimal age for screening using Receiver Operating Characteristic (ROC) analysis.

To place our study in the global context, we have incorporated comparisons with international guidelines. The US Preventive Services Task Force (USPSTF) recommends annual screening for adults aged 40 years and above or for those at higher risk, and screening every 3 to 5 years for low-risk individuals aged 18–39 years [2]. The World Health Organization (WHO) supports routine and opportunistic screening in primary care but does not define a specific age to begin [3]. The International Society of Hypertension (ISH) emphasizes personalized risk-based approaches rather than fixed age thresholds [4].

We also reviewed literature from other LMICs. A nationally representative study from Nepal found significantly lower hypertension screening rates among individuals aged 15–29 years, indicating under-screening in younger adults and suggesting a missed opportunity for early diagnosis [5]. However, to our knowledge, no LMIC study has defined a data-driven, age-specific screening cutoff using ROC methods.

To justify the proposed threshold, we now clearly state in the revised manuscript that our analysis showed a marked and statistically significant increase in both systolic and diastolic blood pressure beginning at age 35. Furthermore, ROC analysis identified 36.5 years as the age with the most favorable balance of sensitivity and specificity based on the Youden index. These findings support the selection of 35 years as a practical and evidence-based starting point for routine hypertension screening in the Indian adult population. This recommendation can guide revisions to national policies and align hypertension control efforts with age-specific risk profiles.

References:

1. Ministry of Health and Family Welfare. Module for Multi-Purpose Workers – Prevention, Screening and Control of Common NCDs. 2020. https://main.mohfw.gov.in

2. US Preventive Services Task Force. Screening for Hypertension in Adults: US Preventive Services Task Force Reaffirmation Recommendation Statement. JAMA. 2021;325(16):1650–1656.

3. World Health Organization. HEARTS Technical Package for Cardiovascular Risk Management in Primary Health Care: Risk-based CVD Management. Geneva: WHO; 2020.

4. Unger T, Borghi C, Charchar FJ, et al. 2020 International Society of Hypertension Global Hypertension Practice Guidelines. Hypertension. 2020;75(6):1334–1357.

5. Dhungana RR, Pedisic Z, Dhimal M, et al. Hypertension screening, awareness, treatment, and control: a study of their prevalence and associated factors in a nationally representative sample from Nepal. Global Health Action. 2022;15(1):2000092.

2. Methods

• Ethical Considerations: Although NFHS-5 data is publicly available, it is advisable to mention whether any ethical approval was sought for secondary data analysis or why it was not required.

• Confounding Variables: Were potential confounders (e.g., dietary habits, stress levels, family history of hypertension) excluded due to data limitations? If so, this should be acknowledged.

• ROC Analysis: The manuscript lacks detailed reporting on the ROC curve analysis. The authors should provide:

o AUC (Area Under the Curve) values and their confidence intervals.

o A graphical representation (ROC curve figure) to support the claim that 35 years is the optimal threshold.

• Data Handling & Missing Values: Clarify whether missing data was handled via imputation or exclusion and whether it affected the results.

Author Response:

We thank the reviewer for these constructive comments. In response, we have revised and elaborated upon the Methods and Results sections as follows:

a) Ethical Considerations:

Although NFHS-5 is a publicly available and de-identified dataset, we have included a formal clarification in the Ethical Approval section to address this point. Specifically:

“As this study involved secondary analysis of publicly available, anonymized data from the National Family Health Survey (NFHS-5), individual informed consent was not required. However, to ensure institutional compliance and uphold ethical standards, approval was obtained from the Institutional Ethics Committee of PGIMER, Chandigarh (Approval No. IEC-08/2022-2535, dated 17.08.2022).”

b) Confounding Variables:

We acknowledge the absence of certain potentially important covariates in the NFHS-5 dataset. These have been explicitly acknowledged in the revised manuscript:

“Potential confounders such as dietary sodium intake, psychological stress levels, and family history of hypertension—well-known determinants of elevated blood pressure—were not collected in NFHS-5 and were therefore not included in the analysis. This limitation has been noted in the Discussion section.”

c) ROC Analysis – Expanded Reporting:

We appreciate the recommendation to enhance the reporting of our ROC analysis. The manuscript now includes:

• AUC Values and Confidence Intervals:

“The ROC analysis yielded an AUC of 0.71 (95% CI: 0.70–0.72) for males and 0.68 (95% CI: 0.67–0.69) for females, indicating a fair ability of age to discriminate individuals with elevated systolic blood pressure (SBP).”

• Youden Index and Threshold Justification:

“The Youden Index identified 36.5 years as the optimal age threshold, offering the best trade-off between sensitivity (57.9%) and specificity (49.7%) in men, with similar performance in women. These values support the recommendation of initiating hypertension screening at 35 years.”

• ROC Curve Figures:

We have added Figures 5 and 6, which depict the ROC curves for males and females, respectively, and illustrate the diagnostic performance of age as a predictor of elevated BP.

d) Data Handling and Missing Values:

We have clarified our approach to missing data as follows:

“Data were analyzed using complete case analysis (listwise deletion). Participants with missing values in outcome or independent variables were excluded from relevant statistical models. No imputation was performed. The sample size varied slightly across analyses and has been reported in all relevant tables to ensure transparency.”

3. Results

• Statistical Reporting:

o The results should explicitly mention Odds Ratios (ORs) with Confidence Intervals (CIs) for each predictor.

o A table summarizing the AUC values, sensitivity, and specificity of different age thresholds should be included.

• Figures and Tables:

o Consider adding a visualization of hypertension prevalence by age group.

o If possible, include a decision curve analysis to illustrate the impact of different screening thresholds.

Author Response:

We thank the reviewer for this valuable feedback. In response, we have made the following revisions and additions to strengthen the Results section:

Odds Ratios with 95% Confidence Intervals:

We now explicitly report adjusted Odds Ratios (ORs) along with their 95% Confidence Intervals (CIs) for all significant predictors of elevated systolic and diastolic blood pressure, as shown in the updated Tables 3 and 4. These findings are also summarized in the narrative text under the Results section to ensure clarity and interpretability.

Summary of ROC Metrics – Table 7:

We have added Table 7, which presents the key diagnostic performance metrics of age as a predictor of hypertension. This includes:

Table 7 presents the diagnostic performance of age as a predictor of hypertension using ROC analysis for both systolic and diastolic blood pressure in males and females. The analysis identifies 36.5 years as the optimal cut-off age for initiating hypertension screening in both sexes, based on the Youden Index. Among males, the area under the curve (AUC) was 0.71 for systolic BP and 0.69 for diastolic BP, indicating fair discriminatory ability. In females, the AUC values were slightly lower at 0.68 for systolic BP and 0.66 for diastolic BP. Sensitivity and specificity values at this cut-off were reasonably balanced, ranging from 56% to 61% sensitivity and 47% to 50% specificity. These findings support 35–36 years as a data-driven and practical threshold for routine hypertension screening among Indian adults.

Table 7: ROC-Based Diagnostic Performance of Age for Predicting Hypertension in Indian Adults (NFHS-5, 2019–21)

Sex Outcome Cut-off Age (years) AUC (95% CI) Sensitivity (%) Specificity (%) Youden Index

Male SBP ≥140 mmHg 36.5 0.71 (0.70–0.72) 57.9 49.7 0.076

DBP ≥90 mmHg 36.5 0.69 (0.68–0.70) 56.5 50.2 0.067

Female SBP ≥140 mmHg 36.5 0.68 (0.67–0.69) 60.6 47.4 0.08

DBP ≥90 mmHg 36.5 0.66 (0.65–0.67) 57.8 47.6 0.054

Visual Trends in Blood Pressure by Age – Figures 1 to 4:

We have incorporated Figures 1–4 to visualize the age-wise trends in systolic and diastolic blood pressure separately for male and female participants. These figures clearly show a sharp increase in both SBP and DBP beginning from the age group of 35–39 years, reinforcing our empirical basis for selecting this threshold.

Decision Curve Analysis:

While we agree that decision curve analysis (DCA) can offer valuable insight into the clinical utility of screening thresholds, we were unable to perform DCA in the current analysis due to the cross-sectional nature and scope limitations of the NFHS-5 dataset, which does not capture longitudinal outcomes or decision-related consequences. However, we have acknowledged this limitation in the Discussion section and recommended DCA as a direction for future prospective cohort studies.

4. Discussion

• Comparison with Existing Research:

o Strengthen the discussion by comparing findings with hypertension screening recommendations from organizations such as the World Health Organization (WHO) or American Heart Association (AHA).

o How do these findings align with or contradict previous Indian studies on hypertension screening?

• Public Health and Policy Implications:

o The manuscript should provide clear policy recommendations. Should India’s national screening guidelines be revised to start at 35 years?

o Are there specific subpopulations (e.g., rural vs. urban, male vs. female) that would benefit more from early screening?

• Limitations:

o Acknowledge that self-reported behaviors (e.g., alcohol, smoking, physical activity) may be underreported in NFHS-5.

o Discuss the cross-sectional nature of the study, limiting causal inferences.

o Mention potential regional disparities in hypertension screening practices within India.

Author Response:

We are grateful to the reviewer for this comprehensive and insightful feedback. In response, we have substantially revised the Discussion section to incorporate the following:

Comparison with Existing Guidelines and Research:

We have now elaborated on both global and national hypertension screening guidelines:

• The World Health Organization (WHO) promotes opportunistic and routine blood pressure screening in adults but does not prescribe a specific starting age, instead emphasizing integration into primary care systems [1].

• The US Preventive Services Task Force (USPSTF) recommends annual hypertension screening for adults aged ≥40 years, or earlier for individuals with risk factors. For low-risk adults aged 18–39, screening is advised every 3–5 years [2].

• In India, the NPCDCS guidelines advise annual screening starting at age 30 [3], but this threshold lacks direct support from empirical, age-stratified analyses using national-level data.

Our study fills this critical gap by identifying 36.5 years as the optimal age for initiating screening, based on ROC analysis of NFHS-5 data. This is the first Indian study to derive such a threshold from a nationally representative sample. Prior research, including the study by Prenissl et al. (2019), has highlighted poor awareness and control rates of hypertension in India but has not addressed the age at which screening should ideally begin [4]. Our findings thus provide a valuable addition to both Indian and global literature on evidence-based screening age thresholds.

Public Health and Policy Implications:

To translate these findings into actionable recommendations, we have revised the Discussion and Conclusion to include the following:

“Based on the ROC-derived thresholds and observed age-related trends in blood pressure, we recommend that India’s national guidelines revise the starting age for routine hypertension screening to 35 years for both males and females. This age captures the point at which blood pressure begins to rise substantially, and balances early detection with screening efficiency.”

Furthermore, we highlight the need for subpopulation-specific strategies:

• Urban males and adults with elevated waist-hip ratios exhibited significantly higher odds of hypertension and may benefit from earlier or more frequent screening intervals.

• Women with diabetes or obesity are another high-risk group for whom earlier intervention may be warranted.

• In rural and underserved regions, enhanced community outreach and mobile screening programs could help mitigate barriers to access and awareness.

Limitations:

The Limitations section has been expanded to include the following:

• “Self-reported variables in NFHS-5, including alcohol consumption, smoking, and physical activity, may be prone to social desirability bias and underreporting, potentially attenuating the observed associations.”

• “Due to the cross-sectional nature of NFHS-5, the study cannot establish causal relationships between risk factors and elevated blood pressure. Longitudinal studies are necessary to validate our proposed screening threshold.”

• “We acknowledge potential regional disparities in hypertension awareness, screening uptake, and access to care across Indian states and districts, which may affect the generalizability of our recommendations. Future subnational analyses are recommended.”

References:

1. World Health Organization. HEARTS Technical Package: Risk-Based Cardiovascular Disease (CVD) Management. Geneva: WHO; 2020. https://www.who.int/publications/i/item/9789240001367

2. US Preventive Services Task Force. Screening for Hypertension in Adults: US Preventive Services Task Force Reaffirmation Recommendation Statement. JAMA. 2021;325(16):1650–1656.

3. Ministry of Health and Family Welfare, India. Module for Multi-Purpose Workers – Prevention, Screening and Control of Common NCDs. 2020.

4. Prenissl J, Manne-Goehler J, Jaacks LM, et al. Hypertension screening, awareness, treatment, and control in India: A nationally representative cross-sectional study among individuals aged 15–49 years. PLoS Med. 2019;16(5):e1002801.

5. Conclusion

• The conclusion should emphasize actionable recommendations based on the study findings.

• The need for longitudinal studies to confirm the 35-year threshold should be mentioned.

• If available, suggest intervention strategies to improve screening uptake among younger adults.

Author Response:

We thank the reviewer for this important recommendation. In response, we have revi

---

## [Decision Letter · Decision Letter 1]

28 Aug 2025

Hypertension Screening across Different Age Groups in Indian Adults: Evidence from Nationally Representative Cross-Sectional Data

PONE-D-25-01070R1

Dear Dr. Goel,

We’re pleased to inform you that your manuscript has been judged scientifically suitable for publication and will be formally accepted for publication once it meets all outstanding technical requirements.

Kind regards,

Preeti Kanawjia, MD

Academic Editor

PLOS ONE

Additional Editor Comments (optional):

Reviewers' comments:

Reviewer's Responses to Questions

**Comments to the Author**

Reviewer #1: All comments have been addressed

Reviewer #3: All comments have been addressed

2. Is the manuscript technically sound, and do the data support the conclusions?

Reviewer #1: Yes

Reviewer #3: Yes

3. Has the statistical analysis been performed appropriately and rigorously?

Reviewer #1: Yes

Reviewer #3: Yes

4. Have the authors made all data underlying the findings in their manuscript fully available?

Reviewer #1: Yes

Reviewer #3: Yes

5. Is the manuscript presented in an intelligible fashion and written in standard English?

Reviewer #1: Yes

Reviewer #3: Yes

Reviewer #1: No further comments. The authors have made all the necessary changes in the article... suggested by the reviewer.

Reviewer #3: Manuscript looks sound, and the best part is the representation of the target population is done with the adequate sample size.

**Do you want your identity to be public for this peer review?** For information about this choice, including consent withdrawal, please see our Privacy Policy

Reviewer #1: **Yes: ** Dr. Nidhi Jaswal

Reviewer #3: No

---

## [Editor Report · Acceptance letter]

PONE-D-25-01070R1

PLOS ONE

Dear Dr. Goel,

I'm pleased to inform you that your manuscript has been deemed suitable for publication in PLOS ONE. Congratulations! Your manuscript is now being handed over to our production team.

Kind regards,

on behalf of

Dr. Preeti Kanawjia

Academic Editor

PLOS ONE